# Music and Tactile Stimuli during Daily Milking Affect the Welfare and Productivity of Dairy Cows

**DOI:** 10.3390/ani13233671

**Published:** 2023-11-27

**Authors:** Karine Keyzy dos Santos Lemes Lechuga, Fabiana Ribeiro Caldara, Maria Fernanda de Castro Burbarelli, Agnês Markiy Odakura, Caio César dos Ouros, Rodrigo Garófallo Garcia, Gisele Aparecida Félix, Ibiara Correia de Lima Almeida Paz, Viviane Maria Oliveira dos Santos, Jaqueline Murback Braz

**Affiliations:** 1School of Agricultural Science (FCA), Federal University of Grande Dourados (UFGD), Dourados 79824-900, MS, Brazil; mvkeyzy@outlook.com (K.K.d.S.L.L.); fariakita@gmail.com (M.F.d.C.B.); m.odakura@hotmail.com (A.M.O.); caio_ouros@hotmail.com (C.C.d.O.); rodrigogarcia@ufgd.edu.br (R.G.G.); braz_jak@hotmail.com (J.M.B.); 2Veterinary Sciences, Grande Dourados University Center (UNIGRAN), Dourados 79824-900, MS, Brazil; giselefelix@unigran.br; 3Department of Animal Production, School of Veterinary Medicine and Animal Science (FMVZ), São Paulo State University (UNESP), Botucatu 18618-687, SP, Brazil; ibiara.paz@unesp.br; 4School of Veterinary Medicine and Animal Science (FAMEZ), Federal University of Mato Grosso do Sul (UFMS), Campo Grande 79070-900, MS, Brazil; viviane.oliveira@ufms.br

**Keywords:** dairy cattle, environmental enrichment, human–animal interaction, serotonin

## Abstract

**Simple Summary:**

Animal health and well-being have become essential criteria for the acceptability of animal-derived products. In dairy cow production, environmental enrichment has become a significant advancement in providing a higher quality of life for these animals, enabling them to cope with the stressful challenges of daily management, as this is one of the production lines that require greater human–animal interaction. Using music and tactile stimuli separately or in combination before the milking period, we discovered that when stimuli are provided, there is a real change in the productivity of the animals. This is supported by data on residual milk quantity and dripping. Additionally, there was an improvement in their quality of life due to an increased level of serotonin, a neurotransmitter responsible for providing a sense of well-being and reducing reactive responses during daily management. The two alternatives for environmental enrichment addressed in this study have shown promise and can be easily incorporated into animal production due to their low installation and maintenance costs. However, more pronounced effects seem to come from the use of music, which also has the advantage of not posing any risk to the animal’s health or milk quality.

**Abstract:**

The objective of this study was to evaluate the effects of tactile stimuli and music during daily milking on the productive, physiological, well-being, and health parameters of dairy cows. The experiment, which lasted 39 days, was conducted on a commercial farm with forty crossbred cows (age: 36 to 42 months; weight: 350 to 400 kg) distributed in a completely randomized design (2 × 2) via the following treatments: (Con)—cows not exposed to stimuli, (Tac)—cows exposed to tactile stimuli before milking, (Mus)—cows exposed to music during milking, (Tac+Mus)—cows exposed to both stimuli. In this study, classical music with a slow (75 to 107 BPM) and moderate tempo (90 to 100 BPM) was played, and tactile stimuli was provided manually using a flexible stick in the posterior region and udders of the cows. Cows not exposed to any of the stimuli had up to 41% higher residual milk quantity compared to those exposed to one of or both of the stimuli. The sound stimulus promoted an increase in milk letdown before the start of milking. Cows exposed to stimuli showed higher serotonin levels, indicating a beneficial effect on animal well-being. However, cows exposed to pre-milking tactile stimuli showed an increase in somatic cell count. Combining both techniques may have positive effects on milk productivity and well-being. However, using music alone may be more recommendable as it does not pose health risks.

## 1. Introduction

In most tropical countries, dairy herds are mainly composed of crossbred cattle (*Bos taurus taurus* x *Bos taurus indicus*) raised in pasture-based systems, making their interactions with humans restricted [1]. Zebu breeds are naturally more reactive than taurines and often exhibit greater aggressive behavior due to their fear of humans, making them more difficult to milk [2,3] as they frequently need to have their hind legs tied for safety reasons, mainly to avoid accidents due to kicks. Sometimes a stick is also used to control such animals. Different interactions with the handler can be used to modulate cattle temperament and can become an important tool when working with a herd dominated by zebu [4].

Routine human behavior and attitudes directly affect the level of animals’ fear of their handlers [5]. Animals in comfortable conditions and exposed to positive experiences in relation to humans tend to feel less fear and, consequently, allow for handling [6]. Considering the strong interaction that exists between handlers and dairy cows during milking, feeding, and healthcare, the importance of using good management practices to achieve better welfare conditions for both becomes evident.

The quality (positive or negative) and the forms of interaction (tactile, visual, olfactory, and auditory) between them may also affect physiology, metabolism, and productivity indexes of the animals since they are directly related to the well-being of cattle and of milkers. Therefore, the introduction of improvements in the environment, making it more suitable for the physical and behavioral needs of the animals, may facilitate this process. Enriching the environment helps animals to cope with stressors in their environment, reduce frustration, and satisfy their behavioral needs, in addition to promoting positive affective feelings [7,8,9].

A positive relationship between humans and animals spontaneously promotes the formation of bonds and affinity between them [10], making animals more predisposed to tactile contact since they understand this interaction as rewarding [11]. This relationship can be achieved through tactile stimuli such as brushing or stroking with the hands, reducing the impact of environmental stressors such as milking apparatus and equipment and human contact [12,13,14]. 

Studies related to the use of manual massages on the udder in the pre-partum period of primiparous cows immediately before the start of milking to stimulate the secretion of oxytocin and improve milking dynamics [15,16,17,18], reduce the animals’ fear of unfamiliar objects and humans [19,20], and use automatic rotating brushes as a form of environmental enrichment in the waiting room [21,22] have demonstrated positive effects on dairy cows’ well-being and productivity, as well as reducing their fear of humans. However, few studies have reported the effects of bodily tactile stimuli (other than the udder) during the milking procedure. Working with Gir cows that received positive tactile stimulation (via hands and manual brushes) in the udder region and hind legs during the pre-partum period, the authors of [23] observed that tactile stimulation carried out in the corral corridor was effective in reducing the reactivity of cows during milking procedures, facilitating the routine use of this management strategy, in addition to reducing the volume of residual milk by 26.39%.

Hence, due to the characteristics of herringbone milking installations, which make closer contact to supply tactile stimuli in different body regions difficult, we propose the hypothesis that simple devices such as flexible and soft sticks can be used to facilitate this practice and promote positive contact between the milker and the cow in the moments before milking.

Several researchers have studied the use of music to reduce the effects of stress in animals kept in intensive production systems [24,25,26,27]. The results of previous studies indicate that playing classical music during milking may have positive effects on milk production and quality and on animal welfare, reducing stress and agitated behavior among dairy cows [28,29]. However, further research is needed to better understand which mechanisms of action may be related to this practice’s beneficial effects. 

Considering these aspects, it is believed that positive tactile and sound stimuli, isolated or together, can favor a reduction in fear and stress during milking, promoting a reduction in adrenaline secretion and a greater release of oxytocin, thus favoring the ejection process of milk. Thus, in the present study, we aimed to evaluate the effects of tactile stimuli and music, used individually or in concert, on the productivity and welfare-related parameters of dairy cows. 

## 2. Materials and Methods

All procedures performed in this study were approved by the Ethics Committee for Animal Use (CEUA) of the Federal University of Grande Dourados—UFGD—under protocol no. 03/2022. The study complied with ARRIVE guidelines for reporting in vivo experiments [30], and all methods were carried out in accordance with the relevant guidelines and regulations.

### 2.1. Location and Characterization of the Property at Which the Management Routine Was Applied

The experiment was conducted between March and April 2022 in a commercial dairy farm located in the municipality of Dourados, MS, in the Midwest region of Brazil. The municipality is located at 22°13′18″ S, 54°48′23″ W, and it is at an altitude of 437 m. During the execution of this study, the average monthly climatic conditions recorded by the meteorological station were as follows: rainfall of 216 mm, a minimum temperature of 19.5 °C, a maximum temperature of 30.5 °C, and a relative humidity (RH) of 76.5%.

The animals were bred in a grazing system, predominantly Brachiaria brizantha pasture, and supplemented with commercial concentrate, receiving minced BRS Capiaçu grass forage (Pennisetum purpureum Schum) after milking. 

Animals’ management on the property, prior to the experiment, consisted of bringing the cows in once a day (5 a.m.) from the paddocks located approximately 300 m from the milking place to the waiting room and, later, to the parlor. 

At the beginning of the study, some adaptations to the management of the cows were necessary (e.g., the milkers attended lectures and received training in order to correct aggressive behavior towards the animals, establishing a rational management protocol at the property). After the training, which was conducted during the animals’ adaptation period (10 days), animals were taken from the paddock to the waiting room in a calm manner, without shouting, respecting their displacement speed, and without using aversive tools (shocks, ropes, or sticks). Workers’ behavior during handling was monitored by researchers throughout the experimental period.

Milking on the property was performed mechanically in a herringbone-shaped room measuring 6.20 m × 5.10 m and without railings on its sides; the room was kept closed except for when the animals entered and exited, and it had the capacity to host ten cows, five on each side of the room.

After entering the milking parlor, the cows were prepared via fulfilling the following steps: the cleaning and disinfection of the teats using 0.1% chlorhexidine and paper towels (pre-dipping), conducting the black background mug test, the subcutaneous administration of 1.0 mL of oxytocin 40 s before the milking of each animal, and the placement of individual mechanical teat cups on the first five cows that had received a dose of oxytocin. After milking, the teat cups were completely removed in order of vacuum loss and post-dipping was performed in a similar way to the pre-dipping protocol. 

The management practices (food, health, etc.) and routine schedules adopted by the property were kept during the experiment, only changing the way the animals were handled, in addition to the daily application of oxytocin, which was suspended immediately after the end of the animals’ period of adaptation.

To monitor the temperature and relative air humidity, two digital thermo-hygrometers with an external probe (Jiaxi—HTC 2A) were installed in the milking parlor at a height of 1.50 m from the floor. The data were recorded after the end of daily evaluations.

### 2.2. Animals and Experimental Design

For the study, 40 dairy cows crossbred between Jersey and Holstein were used, with their ages and body weights ranging between 36 and 42 months and 350 to 400 kg, with an average 162 ± 63.12 days in milk and milk production of 7.4 ± 2.7 L/day (of this total, 3.38 residual liters were obtained after the administration of oxytocin). The experiment lasted 39 days, with the first ten days being dedicated to the workers’ training and the adaptation of the animals to the presence of researchers; during this period, tactile stimuli and music were not implemented. 

During the adaptation period, milk production data were collected daily without the application of oxytocin, and after milking, the residual milk was measured after using oxytocin. 

The animals, individually identifiable via numbered collars, were distributed in a completely randomized design in a 2 × 2 factorial scheme and sorted into the following treatment groups: 

T1—Control (Con): cows not exposed to tactile stimuli or music;

T2—Tactile stimuli (Tac): cows exposed to tactile stimuli before milking;

T3—Music (Mus): cows exposed to music during milking;

T4—Tactile stimuli + Music (Tac + Mus): cows exposed to tactile stimuli and music.

The cows were led from the paddocks to the milking room in two groups at a time. First, cows from the (Con) and (Tac) treatment groups were milked. After milking, these groups returned to the paddocks and cows from the (Mus) and (Tac + Mus) treatment groups were then taken to the waiting room in order to prevent the cows from the (Con) and (Tac) treatment groups from having access to sound stimuli. Maintaining the same management strategy previously carried out on the farm, after the end of each milking session, the cows were removed from the enclosure and returned to the paddock with free access to water and feeder area, where they received supplementation in a collective trough. 

### 2.3. Tactile Stimuli

For the provision of the tactile stimuli, a 50 cm flexible rod was used. The tip was covered with cotton fabric, and the cows were rubbed for 07 s across the back and hind limbs of the animal in the neck–tail direction and for 30 s in the udder region; without touching the cows’ teats, circular movements were performed using the rod, applying gentle but firm pressure, in accordance with a methodology adapted from [31]. After this, the teat cups were attached to start milking.

### 2.4. Sound Stimuli (Music)

For the musical stimuli, a playlist was prepared with suites, symphonies, and concertos at an andante (75 to 107 BPM) and andante moderato tempo (90 to 100 BPM) [32], composed for cello, violin, and string instruments (Bach—Suite no. 1 for Cello in G major, Bach—Violin Concerto in A minor, Mendelssohn—String Symphony no. 4, Bach—Brandenburg Concerto no. 4 in G, Bach—Brandenburg Concerto no. 1 in F, Bach—English Suites, Mozart—Symphony no. 33 in B flat major).

The music was played immediately before the cows entered the milking parlor using two speakers with a RMS power of 500 W, not exceeding 75 DB, as recommended by the authors of [33]. The volume of the music was measured using a decibel meter (KTW Apps) to ensure that the sound was evenly distributed and that it did not exceed the recommended volume. 

### 2.5. Dairy Production, Residual Milk, and Milking Time

After the adaptation period, milk production was measured once a week in a single milking shift (5 a.m.). The measurements were carried out individually. The milk was stored in drums with a capacity of 45 L attached to each milking unit and subsequently measured using a ruler to measure the volume of milk to determine the daily production in liters.

After milking, 1.0 mL of oxytocin was applied subcutaneously on the neck to recover residual milk, and 40 s after hormone administration, the cows were milked again. The administration of oxytocin was carried out only on the days where milk production was evaluated. 

The duration of milking was individually evaluated daily using a stopwatch, recording the exact time of attachment of the teat cups and the time of their removal, standardized via loss of vacuum after the extraction of the last teat cup.

### 2.6. Presence of Subclinical Mastitis (CMT Test)

Concomitantly with the assessment of milk production data, a physical examination of the mammary gland was carried out. After the udders were cleaned and disinfected, samples were collected from each mammary quarter to carry out the test diagnosis of subclinical mastitis (California Mastitis Test—CMT) following the manufacturer’s recommendations (Tadabras^®^). A score of 1 indicates a completely negative reaction, and scores 2–5 indicate increasing degrees of udder inflammatory response and are normally considered indicative of subclinical mastitis.

### 2.7. Respiratory Rate, Ocular Temperature, and Udder Surface Temperature

Respiratory rate during milking was measured once a week by counting flank movements for one minute. For the weekly capturing of images and evaluation of ocular and udder temperature, a thermographic camera (FLIR Systems Inc., Wilsonville, OR, USA) was used. All images were processed using FLIR Report Studio software^®^ version 6.4.18039.1003. The emissivity used was 0.98 (emissivity value for mammalian skin) [34,35,36].

Eye and udder images were always taken from the same side at a 90° angle and at a distance between 0.5 and 1.0 m away from the animal, respectively. Based on the recorded images of the udder, the mean surface temperature (MST) was calculated by selecting 30 points in the image area (Figure 1A). The maximum temperature (°C) within the medial posterior palpebral border area of the lower eyelid and lacrimal caruncle [37] was recorded (Figure 1B).

### 2.8. Behavior in the Milking Parlor

The behavior of the cows was continuously evaluated once a week via the focal animal method from when they entered the milking parlor until teat cup removal according to the methodology proposed by the authors of [38].

All assessments were recorded by the same trained observer by counting the number of times each animal showed the behaviors listed in a pre-established ethogram (adapted from [39]). 

### 2.9. Forced Human Approach Test

The test was always performed by the same evaluator once a week during the experimental period. It was conducted in the paddocks and based on the evaluation of the animal’s reaction to the approach of a person, as described by [39]. The evaluator approached each cow calmly, approaching the animal from the front, walking slowly (one step per second), and keeping their arms and hands close to the body. Using a digital tape, the evaluator measured the allowed distance upon approach in meters before the animal moved away. The test ended when the cow moved away, taking well-defined steps. 

### 2.10. Serum Serotonin Levels

At the end of the adaptation period (10th day) and after the data collection period (39th day), at 6:00 am, 15 mL of blood were collected from all animals by puncturing the coccygeal vein to evaluate their serum serotonin levels. The blood was transferred to tubes containing a clot activator and centrifuged in a digital centrifuge (K14-0815C model KASVI) for 10 min at 3000 rpm and room temperature. The serum was transferred to Eppendorf tubes and frozen at −20 °C until analysis. To evaluate serotonin levels, the samples were analyzed using a commercial enzyme immunoassay kit (Alpco Diagnostics, Windham, NH, USA), with values being expressed in ng/mL. 

### 2.11. Statistical Analyses

Data on milking time, milk production, residual milk, flight distance during the forced approach test, respiratory rate, ocular and udder temperature, and serum serotonin levels were evaluated for normality of residues via the Shapiro–Wilk test and for the homogeneity of variances via Levene’s test. 

As these data are influenced by the conditions and initial characteristics of the herd, the following covariates were added to the mathematical model: initial milk production, initial residual milk, days in lactation, live weight in lactation, order of lactation, initial CMT, final CMT, racial group, milking time at the beginning of the experiment, milking time at the end of the experiment, and collection days for data correction. These selected features were included or excluded according to the best model fit. Because the ocular and udder temperatures were influenced by the environmental temperature at the time of the test, the environmental temperature was added to the mathematical model, in addition to the covariates described above, for data correction. 

An analysis of variance was performed using PROC MIXED of SAS^®^ (2014) to evaluate the effects of and the interaction between the music and tactile stimuli. When interactions were significant, Tukey’s test was used to compare means. When only the main effects were observed, the F test was used to compare means.

Statistical analyses of the CMT results and cow behavior were performed using the SAS GLIMMIX procedure (SAS, version 9.4, SAS Institute Inc., Cary, NC, USA). As they did not meet the assumption of the normality of residues, they were transformed using the LOGNORMAL matrix. As they were influenced by the conditions and initial characteristics of the herd, other covariates were included in the mathematical model for data correction according to the best model fit. To compare means, using the least squares test, the estimates obtained were fitted using the inverse link (lines pdiff ilink) of the GLIMMIX procedure. When significant, means were compared using Tukey’s test. 

Also, the data for serotonin levels measured before the beginning of the experimental period were considered as data paired with the data of each treatment during the experiment and evaluated using the T test. The level of significance for all evaluations was 5%.

## 3. Results

### 3.1. Dairy Production, Residual Milk, and Milking Time

Cows exposed to tactile stimuli and music, either alone or in concert, had a daily milk production approximately 50 to 65% higher than those in the control group. These results are consistent with and inversely proportional to the greater milk retention in the mammary glands of cows that did not receive any stimulus during milking (Table 1). For cows exposed to one of or both forms of stimuli, residual milk represented 23.80 to 24.60% of total milk, while for the non-stimulated cows, this value was approximately 46.40%. 

Music did not affect milking time. However, cows exposed to tactile stimuli showed a milking duration about 13.9% longer than those that did not receive tactile stimuli (Table 1).

### 3.2. Presence of Subclinical Mastitis (CMT Test)

There was no effect of music on subclinical mastitis scores. However, the cows exposed to pre-milking tactile stimuli had higher scores (evaluated using the CMT test) compared to those that did not receive this form of stimulation (Table 2). It should be noted that for all treatments, the scores remained within the classifications of weak reaction (score 1) and distinct positive reaction (score 2).

### 3.3. Respiratory Rate, Ocular Temperature, and Udder Surface Temperature

Regardless of the sound stimulus, the cows that were exposed tactile stimuli immediately before the start of milking showed a higher respiratory rate than those that did not receive the same stimulus (Table 3). 

Using music and tactile stimuli in concert promoted an increase in ocular temperature compared to the cows solely exposed to tactile stimuli. It did not differ, however, from those that only heard music, denoting a clear effect of the sound stimulus on this variable. Environmental enrichment factors had no effect on udder surface temperature.

### 3.4. Behavior of Cows in the Milking Parlor

There was no effect of the treatments on urination, hoofing, negative social interactions, and scratching. The behaviors of stereotyping, vocalizing, and positive social interactions had a very low frequency, and their statistical evaluation was not possible. Cows exposed to the tactile stimuli immediately before milking had a higher frequency of defecation, regardless of the presence or absence of music. In turn, cows exposed to music showed a higher frequency of milk dripping before the start of milking compared to those that were not exposed to sound stimuli, regardless of the use of tactile stimuli (Table 4).

### 3.5. Forced Human Approach Test

Cows exposed to both stimuli at the same time showed a shorter flight distance during the forced approach test compared to those only exposed to tactile stimuli or those only exposed to music. Cows not exposed to music did not differ from each other regardless of whether they were exposed to pre-milking tactile stimuli. The results in Table 5 elucidate the interaction between the two stimuli in reducing the animal’s fear of humans.

The flight distance was evaluated weekly observing the effect of time and its interactions with musical and tactile stimuli on this variable. There was an interaction between weeks of evaluation and the use of music (*p* = 0.0026). Thus, this interaction was unfolded to verify the possible effects of sound stimuli on reducing the fear of the animals over time. Regardless of the use of music, there was a reduction in the flight distance over the experimental weeks, showing that only a change to rational handling through training was effective in reducing the animals’ fear of humans. However, this reduction was significantly faster for animals exposed to sound stimuli (Figure 2).

### 3.6. Serum Levels of Serotonin (5-HT)

There was an isolated effect of both music and tactile stimuli on serotonin levels. Cows exposed to pre-milking tactile stimuli had higher levels of 5-HT compared to those that were not and those that listened to music compared to those that were not exposed to sound stimuli (Table 6).

Through comparing the above-mentioned results with the initial serotonin levels, i.e., those analyzed prior to the stimuli being provided (186.2 ng/mL), it was observed that they were very similar to those obtained in the control treatment at the end of the experiment (213.6 ng/mL), while the other treatments showed a significant increase (*p* < 0.0001), especially when both stimuli were applied concomitantly (Figure 3).

## 4. Discussion

The use of tactile stimuli resulted in longer times to perform milking, contrasting with the effect of music, which did not interfere with the time required to carry out the process, even when milk production was higher, which could indicate an improvement in milk flow. The greater presence of milk dripping before the start of milking observed in the cows exposed to treatments with music corroborated these responses. Delayed milk ejection at the start of milking after removing milk from the cistern (bimodal flow) has been associated with health problems [40], especially of the udder [41,42], which could be related to the lower milk flow of the cows exposed to tactile stimuli since they had higher subclinical mastitis scores. Considering that the complete milking time of cows is also a key factor in maximizing the use of facilities and labor, the use of music alone seems to be more advantageous, as it provides positive results both in terms of increasing milk production and reducing the time required for milking.

Adequate milk ejection requires adequate concentrations of oxytocin, which can be promoted by a series of procedures performed before and during milking [43]. In this sense, the tactile stimulation of the udder with the aim of stimulating neural receptors [18,44,45,46], combined with stimulation in the dorsal and posterior region of the animals, aiming to desensitize them to touch and make proximity to humans and handling milking a pleasurable experience, may have favored the secretion and action of OT, a hormone responsible not only for promoting alveolar contractions that is also related to social behavior and positive mental states [47]. These mechanisms of action are corroborated by the lower milk retention rate, higher serotonin levels, and shorter escape distance observed in the cows exposed to tactile stimuli.

The released OT is transported through the bloodstream to its membrane receptors on myoepithelial cells. These receptors are the same as the ones used by adrenaline, which is normally released in stressful situations [48]. Therefore, both neurotransmitters compete for binding sites, which could reduce oxytocin binding and, consequently, milk ejection in animals subject to fear or stress during milking. Furthermore, adrenaline, by promoting peripheral vasoconstriction, reduces blood flow and, consequently, the arrival of OT to receptors in the mammary gland [49], in addition to preventing its release via neurohypophysis [50]. Therefore, we can infer that the reduction in fear and stress during milking promoted by the environmental enrichment factors tested may have contributed to the reduction in the secretion of this catecholamine, contributing to the better action of oxytocin.

To reduce milk retention problems in the mammary glands, an alternative widely used by producers is the application of exogenous oxytocin, aiming to increase the levels of this hormone in the bloodstream, weakening or breaking the adrenaline connection, and allowing its connection to the receptors of the alveolar glands [51,52]. Despite being a great option in the short term (increasing ejection and reducing milking time), the use of exogenous oxytocin without proper monitoring recurrently has negative effects on productivity, exerts effects on the nutritional characteristics of milk, and increases production costs [53].

One of the hypotheses for the lower milk retention among the cows exposed to tactile and sound stimuli, in isolation or in concert, is that such enrichment methods promoted increased well-being associated with a reduction in fear and stress during handling related to milking. Therefore, there was probably a greater secretion of oxytocin, combined with a reduction in adrenaline secretion, which favored the milk ejection process. Such a hypothesis (that the stimulated cows were in a better condition of well-being and had less fear in relation to humans) may be supported by the higher serum levels of serotonin at the end of the experimental period and smaller flight distances observed during the approach test. According to the authors of [54], manual brushing and massaging caused instantaneous increases in serum OT concentration, which could corroborate the results of the present study, in which the lowest milk retention was found in the group of cows that were stimulated via touch. 

However, in the present study, higher respiratory rates and defecation during milking were identified in cows exposed to tactile stimuli, and these factors are characteristic of responses in situations of fear and stress. These data may be associated with the fact that, at the beginning of the experiment, the cows were not familiar with the management strategy proposed in the present study. The greater expression of these behaviors in the first experimental week may have contributed to the increase in mean values calculated for the four assessment weeks. According to [11], it is important to consider that some animals, especially adults, may have difficulty adapting to touch due to previous negative experiences that generated fear. 

Music can have widespread beneficial effects, but the underlying mechanisms of these effects are not completely understood, especially in animals. Therefore, some of the hypotheses raised in this study are based on studies conducted with human beings. In humans, listening to music activates a multitude of structures in the limbic (hippocampus, amygdala) and paralimbic (caudal orbitofrontal cortex, insula, temporal pole, parahippocampal gyrus) brain regions. It also causes neurochemical changes related to reward, motivation, pleasure, stress, and excitement [55], in addition to facilitating aspects of learning and memory [56,57]. 

Among the regions and circuits within the CNS activated by music are the same limbic structures that respond to oxytocin [58,59]. Given that oxytocinergic systems affect social behaviors, including bonds, social memory, and trust [60], this neurotransmitter could therefore be one of the mediators of the effects of music on the body. The cows of all treatment groups reduced their flight distance over the course of the experiment. However, this reduction occurred more quickly in cows that were stimulated via music, which could be related to a greater ease or speed in recognizing the new gentle handling adopted at the beginning of the experiment, as well as the experimenters, reducing early fear in relation to the others. Therefore, our hypothesis is that these effects may have been mediated by a greater secretion of oxytocin promoted by music.

In addition to improve production and milk flow, cows that were exposed to music individually or in concert with tactile stimuli had higher ocular temperatures. Listening to music may promote autonomic responses of the parasympathetic nervous system (PNS), causing physiological changes in blood circulation, breathing, skin conductivity, body temperature, and heart rate [55,61,62]. The activities of the parasympathetic nervous system are facilitated by oxytocin [63]. It is thus possible to infer that the activation of the PNS could be linked to the changes in ocular temperature observed here. 

Positive emotions mediated via the activation of the PNS [64], which controls the functions of “rest and digestion,” trigger a reduction in heart rate, bronchoconstriction, and vasodilation [65]. Previous studies have shown that music increases parasympathetic activity [66,67], which promotes the vasodilation of capillaries and, consequently, an increase in ocular temperature. In alignment with this, the authors of [68] studied the effects of positive stimulation in dogs through offering palatable treats and observed that their ocular temperatures were higher compared to the initial measurements and that their behavior was consistent with positive activation. 

Although ocular temperature can be a useful indicator in the assessment of emotional states, it is recommended that its interpretation be accompanied by behavioral indexes or supplementary physiological parameters, as in the present study, in which the increase in temperature was associated with the increase in serotonin, which may thus be considered as a positive effect.

Most scientific works that have used music to enrich the environment of production animals, and more specifically for dairy cows, have not described the reasons or which criteria were used for choosing the genre(s) of the music played. The combination of musical characteristics such as main tone, frequency, and rhythm of the music differently affected the levels of the neurotransmitters Glutamate and GABA in rats in [69]. Considering that the glutamatergic and gabaminergic systems are related to oxytocin secretion [70,71], an indirect action of possible increases in glutamate levels may also be involved with an increase in oxytocin secretion. 

Cows exposed to tactile stimuli and/or music had higher serum levels of serotonin (5-HT). During early lactation, the great demand for calcium by the mammary gland exceeds the amount of extracellular reserves and their replacement, requiring a dynamic change in its metabolism [72]. Blood calcium levels are tightly regulated by a homeostatic process controlled by vitamin D, proteins, and hormones, one of which is serotonin, in a negative feedback loop [73]. Increased concentrations of 5-HT induce increased expression levels of the main Ca transporters (ORAI1 and PMCA2), increasing their excretion in milk and causing transient hypocalcemia [74,75,76].

Serotonin can directly stimulate mammary gland-derived parathyroid hormone-related protein (PTHrP), which, after its release into the blood, can then bind to bone receptors and act to release Ca stores to recover its blood concentrations [77]. Therefore, in addition to the benefits related to animal welfare, the stimuli used in the present research can help to modulate calcium metabolism during critical phases, such as the beginning of lactation. Furthermore, recent research has shown that manipulations of Ca and 5-HT metabolism can result in positive effects, even in subsequent lactation [78,79].

It should be noted that the results observed refer to a study evaluating low-producing crossbred cows, which had a high milk retention rate after a single daily milking. New research must be carried out to evaluate the effects of the tested stimuli in technological and high-productivity systems.

## 5. Conclusions

Exposure to tactile stimuli and music during milking promoted improvements in the productivity of dairy cows by reducing milk retention in the mammary glands, indicating the potential of decreasing dependence on the use of exogenous oxytocin. Both stimuli were also efficient in reducing fear towards humans. However, tactile stimuli can trigger unwanted effects, such as an increase in subclinical mastitis.

## Figures and Tables

**Figure 1 animals-13-03671-f001:**
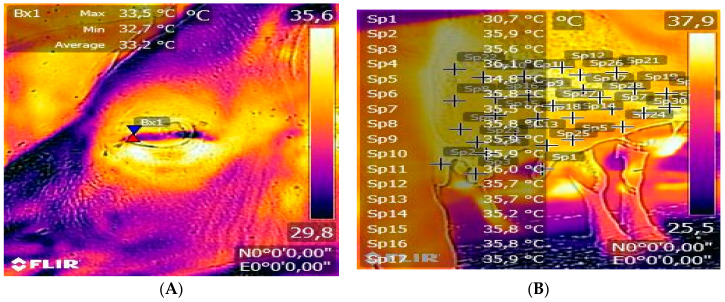
Thermographic image of an udder with marked points for determining the mean surface temperature (**A**) and iris thermographic image with selected maximum temperature point (**B**).

**Figure 2 animals-13-03671-f002:**
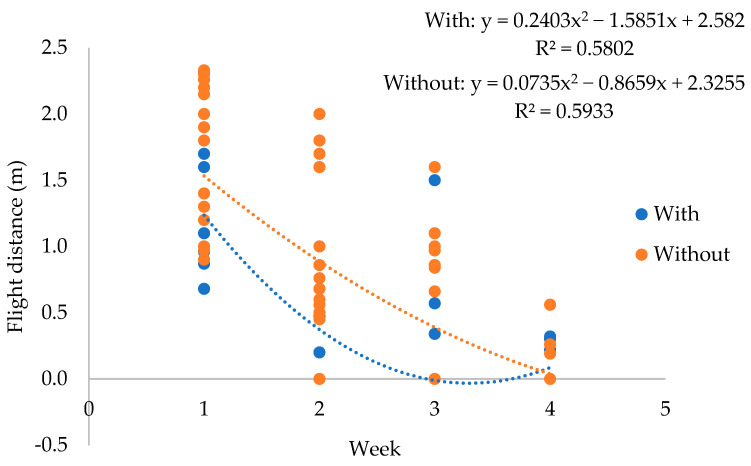
Unfolding of the interaction (*p* = 0.0026) between effects of music over time on the flight distance (m) of the animals.

**Figure 3 animals-13-03671-f003:**
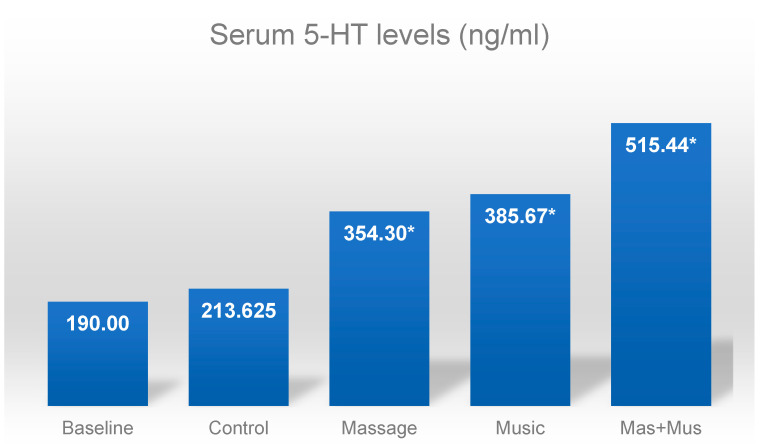
Serum serotonin levels (ng/mL) at baseline and four weeks after exposure to tactile stimuli and/or music in cows during milking. Values followed by * differ from the control (according to a paired data *t* test; *p* < 0.0001).

**Table 1 animals-13-03671-t001:** Milk production, residual milk, and milking time of cows exposed or not to tactile stimuli and/or music during milking.

Variable		Tactile Stimuli		Mean	MSE	*p*-Value		
	Music	With	Without			Tactile Stimuli	Music	Tac × Mus
Milk production (L/day)	With	5.78 Aa	5.63 Aa	5.71	0.21	0.0086	0.0030	0.0316
	Without	5.24 Aa	3.49 Bb	4.36				
	Mean	5.51	4.52	5.02				
Residual milk (L/day)	With	1.84 Aa	1.76 Aa	1.80	0.11	0.0096	0.0209	0.0025
	Without	1.71 Aa	3.02 Bb	2.38				
	Mean	1.86	2.54	2.09				
Milking time (s)	With	705.82	654.53	680.33	1.650	0.0060	0.8341	0.6220
	Without	760.70	633.07	696.88				
	Mean	733.26 a	643.73 b	688.63				

Uppercase letters in columns and lowercase letters in rows differ at the 5% level of significance. MSE—Mean standard error.

**Table 2 animals-13-03671-t002:** Subclinical mastitis scores of cows exposed or not to tactile stimuli and/or music during milking.

Variable		Tactile Stimuli		Mean	MSE	*p*-Value		
	Music	With	Without			Tactile Stimuli	Music	Tac × Mus
CMT	With	1.47	0.98	1.23	0.14	0.0065	0.103	0.422
	Without	2.05	1.18	1.61				
	Mean	1.76	1.08	1.42				

MSE—Mean standard error.

**Table 3 animals-13-03671-t003:** Respiratory rate, ocular temperature, and udder surface temperature values of cows exposed or not to desensitizing tactile stimuli and/or music during milking.

Variable		Tactile Stimuli		Mean	MSE	*p*-Value		
	Music	With	Without			Tactile Stimuli	Music	Tac × Mus
Respiratory rate (BPM)	With	30.29	29.23	29.76	0.27	0.0003	0.3984	0.0651
	Without	30.80	27.72	29.26				
	Mean	30.29 a	28.48 b	29.51				
Ocular temperature (°C)	With	34.16 Aa	33.03 Aa	33.60	0.19	0.1111	<0.0001	0.0144
	Without	31.30 Ba	31.63 Aa	31.46				
	Mean	32.72	32.33	32.53				
Udder Temperature (°C)	With	32.69	32.67	32.68	0.2	0.2147	0.7277	0.2689
	Without	33.39	32.30	32.84				
	Mean	33.04	32.48	32.76				

Uppercase letters in columns and lowercase letters in rows differ at the 5% level of significance. BPM—breaths per minute. MSE—Mean standard error.

**Table 4 animals-13-03671-t004:** Behavioral frequency of cows exposed or not to desensitizing tactile stimuli and/or music during milking.

Variable		Tactile Stimuli		Mean	MSE	*p*-Value		
	Music	With	Without			Tactile Stimuli	Music	Tac × Mus
Urinate	With	0.280	0.310	0.30	0.055	0.2923	0.1649	0.5321
	Without	0.670	0.620	0.65				
	Mean	0.481	0.474	0.48				
Defecate	With	0.530	0.289	0.42	0.045	0.0233	0.7146	0.3313
	Without	0.425	0.550	0.49				
	Mean	0.481 a	0.423 b	0.45				
Vocalize	With	0.153	0.157	0.16	0.034	-	-	-
	Without	0.000	0.300	0.15				
	Mean	0.070	0.230	0.15				
Dripping	With	0.282	0.570	0.428 A	0.04	0.6029	0.0497	0.591
	Without	0.320	0.350	0.337 B				
	Mean	0.303	0.461	0.38				
Stereotypes	With	0.000	0.157	0.08	0.031	-	-	-
	Without	0.000	0.200	0.10				
	Mean	0.000	0.179	0.09				
Hoof	With	0.410	0.078	0.25	0.064	0.1164	0.152	0.7263
	Without	0.300	0.420	0.36				
	Mean	0.354	0.250	0.31				
Negative social interactions	With	0.690	0.105	0.40	0.13	0.4216	0.9418	0.5448
	Without	0.150	0.125	0.14				
	Mean	0.417	0.115	0.27				
Positive social interactions	With	0.131	0.105	0.12	0.03	-	-	-
	Without	0.000	0.070	0.04				
	Mean	0.064	0.890	0.08				
Scratch	With	0.342	0.078	0.21	0.064	0.6891	0.4723	0.2411
	Without	0.400	0.400	0.40				
	Mean	0.371	0.243	0.31				

Uppercase letters in columns and lowercase letters in rows differ at the 5% level of significance. MSE—Mean standard error.

**Table 5 animals-13-03671-t005:** Flight distance (m) during forced approach test of cows exposed or not to desensitizing tactile stimuli and/or music during milking.

Variable		Tactile Stimuli		Mean	MSE	*p*-Value		
	Music	With	Without			Tactile Stimuli	Music	Tac × Mus
Flight distance (m)	with	0.249 Bb	0.593 Aa	0.42	0.06	0.309	0.0001	0.0004
	without	0.808 Aa	0.615 Aa	0.71				
	Mean	0.52	0.60	0.57				

Uppercase letters in columns and lowercase letters in rows differ at the 5% level of significance. MSE—Mean standard error.

**Table 6 animals-13-03671-t006:** Serum serotonin levels (ng/mL) of cows exposed or not to tactile stimuli and/or music during milking.

Variable		Tactile Stimuli		MEAN	MSE	*p*-Value		
	Music	With	Without			Tactile Stimuli	Music	Tac × Mus
Serotonin (ng/mL)	with	515.44	383.75	449.60 A	18.60	<0.0001	<0.0001	0.6219
	without	354.30	213.62	283.96 B				
	Mean	434.87 a	298.69 b	366.77				

Uppercase letters in columns and lowercase letters in rows differ at the 5% level of significance. MSE—Mean standard error.

## Data Availability

The datasets used and/or analyzed for the current study are available from the corresponding author upon reasonable request.

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
