# Peer review of "Music and Tactile Stimuli during Daily Milking Affect the Welfare and Productivity of Dairy Cows"

_animals, 2023, doi:10.3390/ani13233671_

Round 1

Reviewer 1 Report

Comments and Suggestions for Authors

General comment

The study conducted by Lechuga et al. focuses on assessing the impact of desensitizing massage (applying a flexible rod) and music on the welfare and productive performance of dairy cows in a once-a-day milking scenario. While the use of music in such contexts is relatively well-researched, the utilization of desensitizing massage with a rod is somewhat unconventional. Furthermore, the absence of tactile stimulation with hands on the teats, which is not part of the milking protocol, appears to be uncommon. I encountered difficulty in comprehending the rationale behind using desensitizing massage instead of traditional teat stimulation with hands. It is essential for the authors to provide a strong justification for this choice to enhance the relevance of this study for the professionals in dairy production. Furthermore, to enhance the article's acceptability to readers, I suggest shifting the study's focus exclusively to music. Update the manuscript to clearly reflect this in the main findings. The emphasis on the benefits of rod application on milking performance of cows appears less logical. Some other suggestions are described below to improve the quality of the manuscript.  

Title

Please add the phrase “once-a-day milking” to the title to explicitly convey the production system as well.

Summary

Lines 21-23: The desensitizing massage (massaging using a rod) did not appear to have a significant positive impact on udder health. Therefore, the statement could be considered misleading.

Considering the overall findings, music appeared more effective compared to the desensitizing massage in terms of their impact. It would be advisable to provide a clear statement regarding this difference.  

Abstract

Line 31: Could you please include a clear definition of 'desensitizing massage' in this line? It seems to involve massaging the udder and limb with a flexible rod.

Line 41: Once more, in terms of udder health, this statement could be misleading. Please mention the increase in somatic cell count associated with desensitizing massage.

Introduction

The introduction lacks discussion about the use of desensitizing tactile stimuli (flexible rod) in different species and its potential applications in cattle. The use of tactile stimulation with a rod may appear unconventional in practical scenarios. Therefore, the researchers must provide a robust justification for this approach in their study.

Material and Methods

Please use past tense in the Materials and Methods section

Line 119: Please add the average days in milk of the cows.

Line 120: What does this mean that no sensory stimuli were used? Additionally, please add if you administered oxytocin during the first 10 days?

Line 126: Please consider presenting the treatment application in a figure or tabular form (2×2 factorial) instead of listing them.

Line 210: Could you please specify the time of blood sampling?

Line 218: Statistical analyses were well described.

 Results and discussion were well presented. However, the authors need to change their focus of the study as described earlier. Additionally, it would be great if the authors could add the limitations of the study, especially regarding the production level and the once-a-day milking, specifying that the results might not be applicable to the larger dairy population.

Conclusion

Please consider revising the conclusion to specify that desensitizing massage may not be highly effective in improving the welfare of cows. If the somatic cell count is increasing with desensitizing massage, the conclusion suggesting that sensory stimuli could improve the welfare of cows may be somewhat misleading.

Comments on the Quality of English Language

The manuscript needs editing to ensure the appropriate use of tenses, particularly in the Materials and Methods section.

Reviewer 2 Report

Comments and Suggestions for Authors

Dear Authors,

Title: If it is a tactile massage between humans and animals - it should be included in the title, as as it stands, it refers to a mechanical massage. Find the correct term for this type of massage.

The abstract should be completely improved. This sentence is not necessary: The interaction between humans and animals, especially during milking management, can directly affect production and animal well-being. Start with the research objective and present the results further. As it is, it doesn't attract the reader's attention.

Introduction

Line 46-51: That's correct, but your research is on massage with equipment and music. Do you have direct contact with human interaction? If not, this paragraph is out of context. Importance of clarity in the article title. 

Line 70: mechanisms underlying - which?

Line 71: how music can be used more effectively in the management of dairy cattle. Does your research answer this? Is it a research hypothesis?

Line 74: Remove this term "well-being" for welfare.

Line 74-75: on the productive, physiological, sanitary, and well- 74 being parameters of dairy cows: These parameters should not be dissociated from welfare, as total welfare is obtained with their association. I recommend reviewing welfare concepts and adjusting the research objective.

Line 34: The municipality is located at 22º13'18" S, 54º48'23" W, and altitude of 437 m.  Can this location information be disclosed? Review the journal's rules.

Line 89-92: Was feeding taken into consideration for the productive performance of the cows? In other words, how did you separate this dietary variable from environmental warming to evaluate production increase?

Line 97:  Training of milkers: How long of training? Was the handling of human beings monitored after training? Who measures this?

Line 113:  The application of oxytocin was sus pended since the beginning of experimental evaluations. How long before? How was the presence of residual oxytocin in the animal's body assessed?

Line 121-122: This sentence is not clear. 

Line 123-124: So the application of oxytocin was intentional?

Line 136-138: Was this management already done prior to the experiment? If yes, make it clear in the methodology. There are studies that prove that cows have greater motivation to access open paddocks than closed paddocks, thus, it could even influence the faster milk letdown to access the open paddock, if they were not adapted.

Table 1. It is not necessary to present this table, as it is a reference from another author, just cite the authorship in the text.

Line 144-147: The according to which methodology? 

Line: 149-158: Was the animals adapted to the music? Present here in this paragraph. 

Line 160: How long does it take to adapt?

Line 165-168: Describe why oxytocin was used.

Line 194-198: According to which methodology?

Table 2 is not necessary, it was adapted from the author (20). Just cite in the text that was followed the ethogram of the author.

Line 261: Was there a hypothesis for this variable?

Line 335-339: These are results.

Line 348-355: This is a discussion of paper by other authors, there is no need to add this to the discussion of your research, discuss your results.

Line 356-359: In relation to what results? This paragraph is lost in the text

Line: 361-373: This seems like methodology and not discussion.

Line 374-389: This is a literature review. What is the relationship with your results?

Line 390-397: I did not find this hypothesis in the introduction of the paper.

Line 398-405: Can we compare this physiological effect between different species? Review this quote.

Line 406-414: That's why the adaptation period takes place. This statement seems contradictory to the objectives of the work and creates insecurity for the reader. Review this statement. 

Line 415-412: There are several scientific studies with farm animals. It is not necessary to point out work with humans.

Line 431-432: Were these levels measured?

Line: 440-447: Review the correct place to mention the hypotheses. It's not in the discussion. Possible research hypotheses are presented in the introduction, not in the discussion. Only when you have results to state that the given hypothesis was achieved.

Line 448-455; Literature review. The construction of this type of paragraph only contributes to the length of the text. Try to be more objective and relate the discussion to the results.

Review all paragraphs of the discussion. It is very extensive and has little relation or mention to the results of this research.

Reviewer 3 Report

Comments and Suggestions for Authors

I find the paper interesting and the discussion well conducted. It would be good to have more detail on the experimental design and method (time) of data collection for the reasons given in the discussion regarding the factors that may have caused the animals to feel anxious rather than relaxed during treatment (especially when the same milkers who used to beat and yell at them are now massaging them).

Line 79: add reference for ARRIVE guidelines

Line 89: “The dairy cows are crossbred between Jersey and Holstein” should be into the chapter “Animals and Experimental Design”

Lines 144 - 146: It is not explained how and with what the massage was performed in the udder region (and what the udder region includes)

Line 148: In this chapter, explain when exactly the music started: when the cows entered the milking parlor, or after all the cows had taken their places in the milking parlor, or after the pre-milking procedures were completed, or at some other time

Line 150: Add reference for tempo. Depending on the data source, Andante and Andante moderato have a different tempo range than that listed in the manuscript.

Line 210: “At the end of the adaptation period (tenth day) and after the data collection period (39th day).” The way of writing numbers of experimental days should be uniformed trough out the text (e.g. 10th and 39th day) - here and in lines 145 and 146.

Line 267: (Table 4) The calculation method of the subclinical mastitis score was not mentioned in the materials and methods, it should be added.

Round 2

Reviewer 1 Report

Comments and Suggestions for Authors

The authors have adequately addressed the concerns raised in the previous feedback. I commend them for their thorough and effective response to the comments. Best wishes for a successful publication.

Author Response

Dear editor and reviewers

The authors would like to thank again the contributions made to the article. To better clarify doubts regarding the proposal to use a stick to perform tactile stimuli, the authors chose to make some changes in the introduction of the article. We hope that we have adequately clarified the questions and remain available for any further improvements that may be necessary. 

Best regards

Reviewer 2 Report

Comments and Suggestions for Authors

Dear authors,

Thank you for the answers to the questions.

Author Response

(The authors gave the same response as above.)
